# Obesity in childhood, socioeconomic status, and completion of 12 or more school years: a prospective cohort study

Louise Lindberg,[1] Martina Persson,[2,3,4] Pernilla Danielsson,[1] Emilia Hagman  ,[1] Claude Marcus[1]

[1]Department of Clinical Science, Intervention and Technology, Karolinska Institutet, Stockholm, Sweden
[2]Department of Medicine Solna, Karolinska Institutet, Stockholm, Sweden
[3]Department of Diabetes and Endocrinology, Sachsska Children's Hospital, Södersjukhuset, Stockholm, Sweden
[4]Department of Clinical Science and Education, Södersjukhuset, Karolinska Institutet, Stockholm, Sweden

**Correspondence to**
Dr Emilia Hagman;
emilia.hagman@ki.se

## ABSTRACT

**Objectives** Children with obesity achieve lower educational level compared with normal-weight peers. Parental socioeconomic status (SES) impacts both a child's academic achievement and risk of obesity. The degree to which the association between obesity and education depends on parental SES is unclear. Therefore, the primary aim is to investigate if individuals with obesity in childhood are less likely to complete ≥12 years of schooling, independently of parental SES. The secondary aim is to study how weight loss, level of education and parental SES are associated.

**Design** Nationwide prospective cohort study.

**Setting** Swedish national register data.

**Participants** Children aged 10–17 years, recorded in the Swedish Childhood Obesity Treatment Register, and aged 20 years or older at follow-up were included (n=3942). A comparison group was matched by sex, year of birth and living area (n=18 728). Parental SES was based on maternal and paternal level of education, income and occupational status.

**Primary outcome measure** Completion of ≥12 years of schooling was analysed with conditional logistic regression, and adjusted for group, migration background, attention deficit disorder with or without hyperactivity, anxiety/depression and parental SES.

**Results** Among those with obesity in childhood, 56.7% completed ≥12 school years compared with 74.4% in the comparison group (p<0.0001). High parental SES compared with low SES was strongly associated with attained level of education in both children with and without obesity, adjusted OR ($^a$OR) (99% CI)=5.40 (4.45 to 6.55). However, obesity in childhood remains a strong risk factor of not completing ≥12 school years, independently of parental SES, $^a$OR=0.57 (0.51 to 0.63). Successful obesity treatment increased the odds of completing ≥12 years in school even when taking parental SES into account, $^a$OR=1.34 (1.04 to 1.72).

**Conclusions** Individuals with obesity in childhood have lower odds of completing ≥12 school years, independently of parental SES. Optimised obesity treatment may improve school results in this group.

## Strengths and limitations of this study

► In this prospective cohort study, we have been able to investigate the level of education among a large number of individuals who have obesity in childhood (n=3942) in comparison with a matched group (n=18 728).
► The study design of using longitudinal data from several national registers provided the opportunity to control for important confounding factors, such as neuropsychiatric disorders, anxiety, depression and family socioeconomic status.
► Factors such as free education, school lunches and students' healthcare may have an impact on the generalisability of our data to other populations.

## INTRODUCTION

During the last 40 years, the prevalence of childhood obesity has increased exponentially in many parts of the world.[1] Childhood obesity is associated with increased risks of somatic morbidity and risk of premature death in adulthood.[2–5] Psychosocial maladjustment, anxiety and depression are also prevalent[6 7] and may contribute to the obesity-related long-term morbidity and mortality.[5] Most studies report that children with obesity more often have lower school grades and reach a lower level of education compared with normal-weight peers,[8–10] but in a recent study, only girls with obesity were affected.[11] We have previously confirmed the association between obesity and lower attained level of education among both girls and boys.[9] We also found that successful obesity treatment was positively correlated with completing ≥12 school years, although without taking socioeconomic status (SES) into account.[9]

Low parental SES is a well-established risk factor for both childhood obesity and poorer academic achievement.[12] In particular, parental education has been demonstrated to influence the child's performance at school.[13–15] In addition, depression[16] and neuropsychiatric disorders, including attention deficit disorder with or without hyperactivity (ADHD/ADD),[17] may adversely affect school results. Depression and

BMJ

neuropsychiatric disorders are more common in children with obesity compared with the general population.[9 18] Investigating the impact of obesity alone on attained level of education requires that these and other confounders are considered. The primary aim of this study was to disentangle the association of childhood obesity and parental SES on completed level of education. The secondary aim was to study if positive effects of weight loss on attained level of education are affected by parental SES.

## METHODS

### Study population

This prospective cohort study included children with obesity,[19] aged 10–17 years at the start of obesity treatment (December 1994–December 2015), and aged 20 years or older and living in Sweden at follow-up (31 December 2017; n=3942). Data on subjects receiving treatment for childhood obesity were collected from the Swedish Childhood Obesity Treatment Register (BORIS). BORIS has been thoroughly described elsewhere,[20] but in short: the main purpose of the register is quality assessment and long-term evaluation of childhood obesity treatment. Treatment is individualised and consists primarily of lifestyle modification (ie, diet and physical activity).

A comparison group from the general population was randomly identified using the Swedish Total Population Register and matched by sex, year of birth and living area at the year obesity treatment was initiated (n=18 728). Using density matching without replacement, five individuals were matched to each individual from the childhood obesity cohort. Siblings of children registered in the childhood obesity cohort were excluded from the comparison group. Children with a diagnosis of mental retardation or genetic syndromes were excluded from both the childhood obesity cohort and the comparison group (figure 1 and online supplemental table 1).

Families were informed in written or verbal about data collection in BORIS. Post an opt-out approval by parents/guardians, data of the children's weight and height were recorded by the local healthcare provider during treatment visits. There were no data of weight and height of individuals in the comparison group. However, less than 1% of the individuals in the comparison group were found in the National Patient Register with a diagnosis of obesity in childhood. In Sweden, healthcare is free of charge for children and adolescents until 18 years of age.

### Data sources

Using the Swedish identity number, which is unique to each resident in Sweden, data from several national registers were linked.[21] Data on education, income and occupational status were obtained from the Longitudinal Integration Database for Health Insurance and Labour Market Studies. Information on migration background for children and their parents was obtained from the Swedish Total Population Register.[22] Both these registers are held by Statistics Sweden, a governmental agency that collects and provides official statistics (www.scb.se/en).

Data on diagnoses of mental retardation, genetic syndromes, ADHD/ADD, anxiety and depression were identified based on codes according to the International Classification of Diseases (10th revision; online supplemental table 1) and retrieved from the National Patient Register.[23] Information on prescriptions of psychostimulant drugs for ADHD/ADD as well as antidepressants and tranquillisers for anxiety and depression was identified using the Anatomical Therapeutic Chemical classification system (online supplemental table 1) retrieved from the Swedish Prescribed Drug Register.[24] Data on deaths were retrieved from the Cause of Death Register.[25] These registers are held and were linked by the governmental agency the National Board of Health and Welfare (www.socialstyrelsen.se/english).

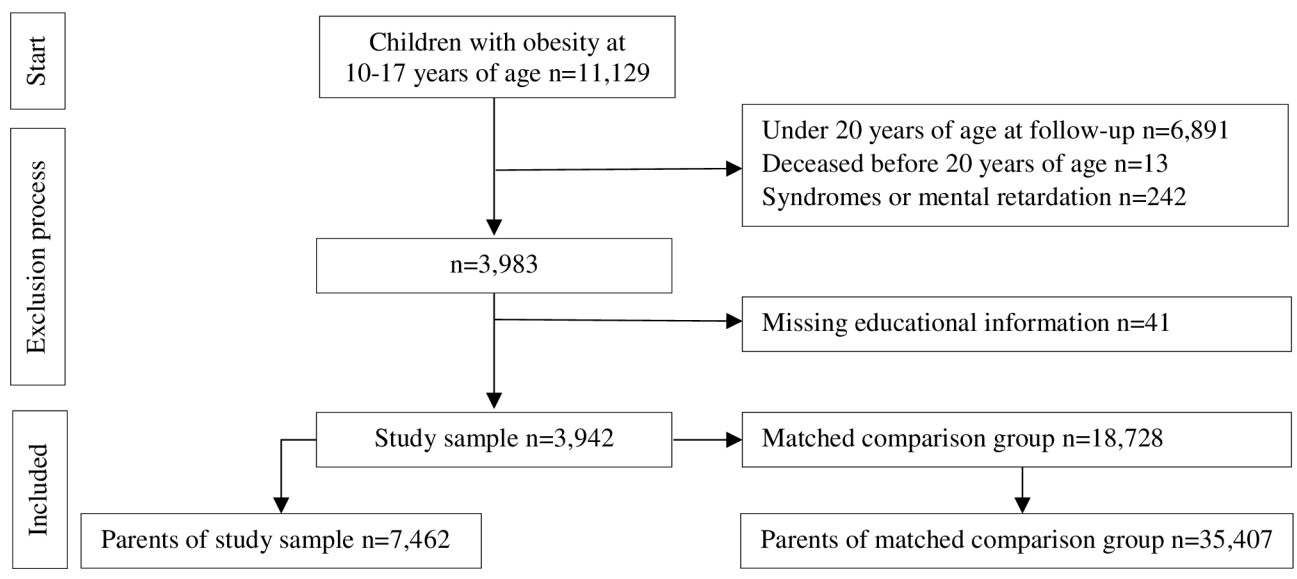

**Figure 1** Participant flow chart.

## Definition of outcome

The main outcome was defined as completion of ≥12 years of schooling, and based on the International Standard Classification of Education.[26] In Sweden, children start school at the age of 6 or 7 years and attend compulsory school for 9 years. Upper secondary school includes 3 additional years of schooling and provides the requirements to attend higher education. Usually students graduate from upper secondary school at 18 or 19 years of age. All education in Sweden is free of charge. Students in compulsory and upper secondary school are provided school lunches and healthcare at schools, also free of charge.

## Definition of exposure variables

Degree of obesity in children from BORIS was assessed with body mass index SD score (BMI SDS), which is standardised by sex and age and used to measure degree of obesity in growing children.[19] Baseline measures (continuous) were used for the variable BMI SDS at start of treatment. Response to obesity treatment was based on the change of BMI SDS from the first to the last clinical visit and categorised into four groups:[3 27] good response, a reduction of BMI SDS by 0.25 units or more; no response, a change of BMI SDS by ±0.24 units; poor response, an increase of BMI SDS by 0.25 units or more; and dropouts, children with less than 1 year between their first and last measure or without clinical follow-up after their first registered visit.

Parental SES was based on maternal and paternal level of education, income and occupational status at the year the child turned 15 years, which is about the same time as the child starts upper secondary school. In case of the child was adopted, the SES of adoptive parents was used (childhood obesity cohort n=24 and comparison group n=164). The rationale of treating SES as a composite variable was to capture more of the social context and a potential inequality embedded there. Thus, by taking three variables into account instead of one, we get a more wide and robust measure of SES. Maternal and paternal educational level was categorised into compulsory school, upper secondary school or university degree. Annual disposable income was used to reflect maternal and paternal economic capacity. The annual disposable income includes all taxable (direct labour income, capital gains from shares and so on) and tax-free income (housing and child benefits, student aid and so on), minus final tax and other negative transfers such as capital loss from shares and properties. Disposable income from different years was converted to 2017 prices using the Consumer Price Index for Sweden provided by Statistics Sweden ( www.scb.se/en). Income was categorised into quartiles based on data from the parents in the comparison group. No occupation was defined as either unemployment 6 months or more, or income from long-term sick leave exceeding any income from the individual's gross salary. Individuals considered to have an occupation included those registered as employed or having an income from

student grants/loan equivalent to full-time studies for at least one semester. Parental SES was weighted according to maternal and paternal education (0, 1, 2), income (0, 1, 2, 3) and occupational status (0, 1). The mean parental SES score (ie, the sum of the mother's and father's all SES indicators divided by two) was applied to their child. The SES variable was categorised into four levels: low SES (0–1.5 points), medium-low SES (2–3 points), medium-high SES (3.5–4.5 points) and high SES (5–6 points).

The prevalence of both ADHD/ADD[9] and depression[18] is higher in children with obesity and may negatively influence attained level of education.[9 16 17] ADHD/ADD and anxiety/depression in children were identified based on diagnosis or dispensed prescribed medication (online supplemental table 1).

## Definition of covariates

The migration background of children in Sweden may impact school achievements and is therefore an important factor to control for in the analyses.[9 14] Migration background was categorised as Nordic or non-Nordic. Nordic was defined as born in a Nordic country (Sweden, Norway, Denmark, Finland or Iceland) with one or two parents born in the Nordic. Children were classified as non-Nordic if born outside the Nordic or born in the Nordic with two parents born outside the Nordic. Other covariates included sex and age (continuous) at start of obesity treatment.

## Patient and public involvement

No patient involved.

## Statistical analysis

Descriptive statistics are presented as means and CIs, medians and IQRs, or frequencies and percentages. Conditional logistic regression was used to calculate ORs with 99% CI for the main outcome, that is, completing ≥12 school years. Conditional logistic regression was used above ordinary logistic regression as the childhood obesity cohort and the comparison group were matched by several variables. Independent variables included in the adjusted analyses were migration background, ADHD/ADD, anxiety/depression, parental SES and group (childhood obesity cohort or comparison group) in analyses including both groups, otherwise stratified by group. Interaction between childhood obesity and parental SES was tested. The association of each SES indicator on the odds of completing ≥12 school years was also analysed separately for mothers and fathers. Sensitivity analyses excluding individuals with a non-Swedish background (defined as child born in Sweden with at least one parent also born in Sweden), or ADHD/ADD, or anxiety/depression were performed.

Secondary analyses were performed within the childhood obesity cohort to examine associations between patient characteristics and completed educational level. In ordinary logistic regressions, odds were adjusted for sex, migration background, parental SES, age and BMI

SDS at start of treatment, treatment response, ADHD/ADD and anxiety/depression. We tested for possible interaction between parental SES and treatment response for the odds of completing ≥12 years of schooling.

As missing data on parental SES were rare in both groups (table 1), records with missing data were excluded from the analyses, that is, data were not imputed. P values of <0.01 were considered statistically significant. All analyses were performed using SAS statistical software (V.9.4).

## RESULTS

In total, 3942 individuals in the childhood obesity cohort and 18 728 individuals in the comparison group were included in the study (table 1). In both groups, 46% of the participants were girls and the median age at follow-up was 23.4 years. The proportions of individuals of Nordic origin were similar in both groups (73.7% vs 75.0%; p=0.02). Despite the groups being matched for living area, a greater proportion of children with obesity grew up in households with low SES compared with the comparison group (22.0% vs 14.4%; p<0.0001).

### Parental SES, childhood obesity and the child's attained level of education

In the childhood obesity cohort, 56.7% completed ≥12 years in school, compared with 74.4% in the comparison group. Girls more frequently completed ≥12 years of schooling than boys in both groups (online supplemental table 2).

Higher parental SES was positively associated with completion of ≥12 years of schooling in both the childhood obesity cohort and in the comparison group (figure 2). The odds were more than five times higher among children growing up in high SES households compared with those growing up in low SES households (table 2). However, even when taking parental SES and other risk factors into account, individuals in the childhood obesity cohort were almost half as likely to complete ≥12 school years compared with those in the comparison group (table 2). The adjusted OR ($^a$OR) (99% CI) to complete ≥12 years of schooling stratified by parental SES, when comparing the childhood obesity cohort with the comparison group, showed a trend towards lower OR in the higher level of SES: low parental SES=0.69 (0.50 to 0.95), p=0.0026; medium-low parental SES=0.59 (0.48 to 0.72), p<0.0001; medium-high parental SES=0.46 (0.35 to 0.60), p<0.0001; high parental SES=0.27 (0.14 to 0.54), p<0.0001. P value for interaction test for childhood obesity and parental SES reaches a p value of 0.0015.

The association of parental SES on school performance was more pronounced in the comparison group than in the childhood obesity cohort (online supplemental table 3). For example, stratified analyses show that $^a$OR of completing ≥12 years in school in low SES compared with high SES in the comparison group was 0.17 (99% CI 0.14 to 0.21, p<0.0001) and in the childhood obesity cohort 0.31 (99% CI 0.22 to 0.45, p<0.0001). Regardless of how

we divide the childhood obesity population (into age or calendar year at start of obesity treatment, SES subscores), large differences between the two groups remain (table 2 and online supplemental table 4).

Of note, having a non-Nordic, compared with a Nordic background, was associated with reduced odds to complete 12 or more years of schooling (table 2). This was however not observed in boys in the childhood obesity cohort (p=0.68). Further, excluding individuals with a non-Swedish background, or ADHD/ADD, or anxiety/depression in sensitivity analyses, did not alter the association between childhood obesity and attained level of education. For example, when excluding individuals with a non-Swedish background, the $^a$OR to complete ≥12 years in school for the obesity cohort versus the comparison group was 0.55 (0.48 to 0.63), p<0.0001.

### Degree of obesity and treatment response on completed educational level in children with obesity

In the childhood obesity cohort, the mean age at start of treatment was 13.5 years (99% CI: 13.45 to 13.62) and the mean BMI SDS was +2.91 (99% CI: 2.89 to 2.92). The median treatment duration was 2.83 years (IQR: 1.86–4.42) and the mean treatment response, calculated using BMI SDS from the first to the last clinical visit, was −0.13 (99% CI: −0.15 to −0.10, n=2709, 68.7%, dropouts excluded). At baseline, 50.7% had obesity and 49.3% had morbid obesity. At last measured weight and height, 33.4% had obesity and 47.2% had morbid obesity, while 19.4% of the children no longer had obesity (dropouts excluded).

A greater degree of obesity at start of treatment lowered the odds of completing ≥12 years of schooling, $^a$OR (99% CI)=0.51 (0.40 to 0.64), p<0.0001, per one unit increase in BMI SDS, while age at start of obesity treatment did not influence the outcome (table 3). Treatment response was categorised as good response (n=847), no response (n=1 315), poor response (n=547) and dropouts (n=1233). Of those with good treatment response, 67% completed ≥12 years in school compared with 58%, 52% and 50% in the groups with no or poor response, and dropouts, respectively (p<0.0001). Within all SES groups, except for high SES, greater treatment response resulted in higher odds of completing ≥12 years in school (figure 3). Dropouts were less likely to complete ≥12 years in school compared with non-respondents (table 3) but this was not observed in the high SES group (figure 3). Duration of treatment differed between children with good and poor treatment response (3.5 vs 4.0 years; p<0.0001). In the childhood obesity cohort, the association between parental SES and odds of completing ≥12 years of schooling was not modified by treatment response (p=0.603).

## DISCUSSION

In this prospective cohort study, we have compared level of education among individuals with obesity recorded in

**Table 1** Characteristics of the participants (n=22 670)

| | Childhood obesity cohort (n=3942) | | Comparison group (n=18 728) | |
|---|---|---|---|---|
| | n | % | n | % |
| Female sex | 1825 | 46.3 | 8701 | 46.5 |
| Nordic | 2905 | 73.7 | 14 048 | 75.0 |
| Age at end of follow-up (years) | 23.4* | 21.4–26.3* | 23.4* | 21.5–26.3* |
| ADHD/ADD | 617 | 15.7 | 1044 | 5.6 |
| Anxiety/depression | 831 | 21.1 | 2158 | 11.5 |
| Parental SES | | | | |
| Low | 867 | 22.0 | 2700 | 14.4 |
| Medium-low | 1540 | 39.1 | 6023 | 32.1 |
| Medium-high | 1159 | 29.4 | 6587 | 35.2 |
| High | 355 | 9.0 | 3218 | 17.2 |
| Missing | 21 | 0.5 | 200 | 1.1 |
| Maternal education | | | | |
| Compulsory school | 1582 | 40.1 | 5615 | 30.0 |
| Upper secondary school | 1541 | 39.1 | 7167 | 38.3 |
| University degree | 730 | 18.5 | 5383 | 28.7 |
| Missing | 89 | 2.3 | 563 | 3.0 |
| Paternal education | | | | |
| Compulsory school | 1273 | 32.3 | 4402 | 23.5 |
| Upper secondary school | 1890 | 48.0 | 9095 | 48.6 |
| University degree | 482 | 12.2 | 3912 | 20.9 |
| Missing | 297 | 7.5 | 1319 | 7.0 |
| Maternal income | | | | |
| Q1 | 1173 | 29.8 | 4887 | 26.1 |
| Q2 | 1281 | 32.5 | 5843 | 31.2 |
| Q3 | 931 | 23.6 | 4585 | 24.5 |
| Q4 | 486 | 12.3 | 2988 | 15.9 |
| Missing | 71 | 1.8 | 425 | 2.3 |
| Paternal income | | | | |
| Q1 | 1127 | 28.6 | 3917 | 20.9 |
| Q2 | 760 | 19.3 | 3128 | 16.7 |
| Q3 | 889 | 22.5 | 4389 | 23.4 |
| Q4 | 864 | 21.9 | 5986 | 32.0 |
| Missing | 302 | 7.7 | 1308 | 7.0 |
| Maternal occupational status | | | | |
| Occupation | 2963 | 75.2 | 15 405 | 82.2 |
| No occupation | 916 | 23.2 | 2956 | 15.8 |
| Missing | 63 | 1.6 | 367 | 2.0 |
| Paternal occupational status | | | | |
| Occupation | 2868 | 72.8 | 14 906 | 79.6 |
| No occupation | 800 | 20.3 | 2627 | 14.0 |
| Missing | 274 | 6.9 | 1195 | 6.4 |

Data are n % if not else stated.
*Median with IQRs.
ADHD/ADD, attention deficit disorder with or without hyperactivity; Q, quartile; SES, socioeconomic status.

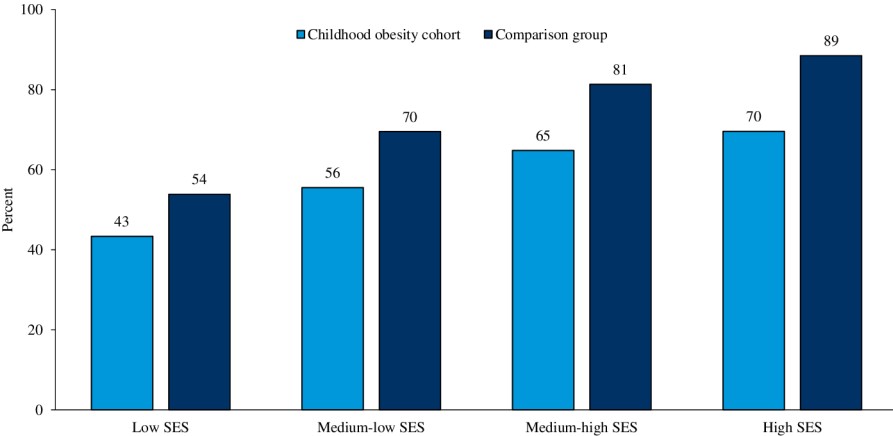

**Figure 2** Percentage of individuals completing ≥12 years of schooling by SES level in the childhood obesity cohort (n=3921) and the comparison group (n=18 528). SES, socioeconomic status.

BORIS, and a comparison group matched by sex, year of birth and living area. Individuals in the childhood obesity cohort were half as likely to complete 12 or more years of schooling, independently of parental SES.

Among individuals from high SES families, those in the childhood obesity cohort were approximately one-third as likely to complete ≥12 years of schooling as individuals from high SES families in the comparison group. Furthermore, our results indicate that parental SES was more important to complete ≥12 school years in the comparison group than in the childhood obesity cohort. An association between obesity and impaired academic achievements has been demonstrated before.[8 9] To which extent other psychosocial factors contribute to this finding has not been evaluated. It is well established that low parental SES,[13–15] immigrant background,[9 14] ADHD/ADD[9 17] and depression[9 14 16] may negatively affect performance in school. These factors are also over-represented in the paediatric population with obesity. In this study, we can confirm that low parental SES, neuropsychiatric disorders and anxiety/depression are more common in children with obesity and contribute to the decreased odds of completing ≥12 school years. However, we found that obesity in childhood is a considerable risk factor for not completing ≥12 years of schooling even after taking these and other important risk factors into account.

### Obesity treatment outcome and educational level
We identified a positive association between weight loss during obesity treatment and completed educational level, which confirms our previously reported data unadjusted for SES.[9] Children with good treatment response, compared with those with no response, were more likely to complete ≥12 school years in all SES groups. A decrease of 0.25 BMI SDS, that has previously been shown to improve metabolic health[27] we can now also show, may have a positive association on completing ≥12 years of schooling. However, the positive effects of successful obesity treatment did not compensate for the observed SES differences on attained level of education.

In Sweden, 12 years of schooling is not mandatory, but the long-term social effects are considerable for those who fail. It has been estimated that 50% of those who fail to complete ≥12 school years face a situation of being left outside the society with poor psychosocial health and large costs for the society.[28] Thus, it is possible that school failure will further worsen the health perspective of adolescents with obesity.

### Potential mechanisms between obesity, weight loss and school performance
Several parameters may interplay when measuring school completion, for example, resourcefulness, intelligence or ability to conform. Nevertheless, the mechanisms by which obesity influences school performance are unclear but are likely multifactorial and complex. Possible mechanisms may include psychosocial aspects such as stigma[29] and increased risk for anxiety and depression.[7] Physiological mechanisms may be mediated by anatomical changes such as atrophy of the frontal lobe and reduced hippocampal volume which have been demonstrated in individuals with obesity.[30] Some of the reported cognitive dysfunctions in children and adolescents with obesity include slower response times when performing visuospatial attention tasks,[31] diminished executive functions, such as working memory,[32] and slower cognitive performance speed.[33] In addition, obesity-related comorbidities such as insulin resistance,[34] type 2 diabetes mellitus,[35] chronic low-grade inflammation[36] and the metabolic syndrome[37] have also been associated with impaired executive function, memory performance, attention and cognitive flexibility. A high-fat diet in rodents demonstrated detrimental effects on memory and executive functions.[30]

At least some of these effects seem to be reversible, which may add to the positive associations we observed of weight loss on school outcome. Weight loss may have a direct positive effect on cognitive functions.[38] Extensive weight loss via bariatric surgery improves insulin sensitivity and decreases systematic inflammation,[39] and it has been suggested that these factors affect cognitive functions.[40]

**Table 2** Crude and adjusted OR (99% CI); p value of subjects completing ≥12 years of schooling

| | Crude estimates | Model 1, n=22 449 | Model 2, n=20 610 | Model 3, n=20 691 | Model 4, n=20 885 | Model 5, n=20 420 |
|---|---|---|---|---|---|---|
| Childhood obesity cohort versus comparison group | 0.44 (0.40 to 0.48); <0.0001 | 0.57 (0.51 to 0.63); <0.0001 | 0.55 (0.50 to 0.62); <0.0001 | 0.54 (0.48 to 0.61); <0.0001 | 0.53 (0.48 to 0.59); <0.0001 | 0.59 (0.52 to 0.66); <0.0001 |
| Non-Nordic versus Nordic | 0.70 (0.63 to 0.78); <0.0001 | 0.85 (0.75 to 0.97); 0.001 | 0.77 (0.67 to 0.88); <0.0001 | 0.87 (0.76 to 0.99); 0.0068 | 0.81 (0.71 to 0.92); <0.0001 | 0.97 (0.84 to 1.12); 0.5687 |
| ADHD/ADD versus non-ADHD/ADD | 0.19 (0.17 to 0.23); <0.0001 | 0.28 (0.24 to 0.33); <0.0001 | 0.27 (0.22 to 0.33); <0.0001 | 0.27 (0.22 to 0.32); <0.0001 | 0.26 (0.22 to 0.31); <0.0001 | 0.28 (0.23 to 0.34); <0.0001 |
| Anxiety/depression versus no anxiety/depression | 0.31 (0.27 to 0.35); <0.0001 | 0.39 (0.34 to 0.45); <0.0001 | 0.36 (0.31 to 0.41); <0.0001 | 0.38 (0.33 to 0.44); <0.0001 | 0.39 (0.33 to 0.44); <0.0001 | 0.36 (0.31 to 0.42); <0.0001 |
| Parental SES | | | | | | |
| Medium-low versus low parental SES | 1.74 (1.54 to 1.97); <0.0001 | 1.69 (1.48 to 1.93); <0.0001 | | | | |
| Medium-high versus low parental SES | 3.33 (2.92 to 3.81); <0.0001 | 2.99 (2.59 to 3.45); <0.0001 | | | | |
| High versus low parental SES | 6.21 (5.19 to 7.44); <0.0001 | 5.40 (4.45 to 6.55); <0.0001 | | | | |
| Maternal education | | | | | | |
| Upper secondary school versus compulsory school | 1.88 (1.69 to 2.08); <0.0001 | | 1.63 (1.45 to 1.83); <0.0001 | | | 1.51 (1.34 to 1.71); <0.0001 |
| University degree versus compulsory school | 3.23 (2.86 to 3.66); <0.0001 | | 2.45 (2.12 to 2.83); <0.0001 | | | 2.17 (1.87 to 2.53); <0.0001 |
| Paternal education | | | | | | |
| Upper secondary school versus compulsory school | 1.88 (1.69 to 2.09); <0.0001 | | 1.58 (1.40 to 1.77); <0.0001 | | | 1.42 (1.26 to 1.60); <0.0001 |
| University degree versus compulsory school | 3.07 (2.66 to 3.55); <0.0001 | | 2.09 (1.77 to 2.45); <0.0001 | | | 1.81 (1.53 to 2.15); <0.0001 |
| Maternal income | | | | | | |
| Q2 versus Q1 | 1.28 (1.15 to 1.43); <0.0001 | | | 1.25 (1.10 to 1.41); <0.0001 | | 1.07 (0.93 to 1.23); 0.2065 |
| Q3 versus Q1 | 1.45 (1.28 to 1.63); <0.0001 | | | 1.35 (1.17 to 1.55); <0.0001 | | 1.05 (0.90 to 1.23); 0.4224 |
| Q4 versus Q1 | 1.86 (1.60 to 2.15); <0.0001 | | | 1.52 (1.28 to 1.81); <0.0001 | | 1.07 (0.88 to 1.28); 0.3831 |
| Paternal income | | | | | | |
| Q2 versus Q1 | 1.36 (1.19 to 1.55); <0.0001 | | | 1.31 (1.13 to 1.51); <0.0001 | | 1.07 (0.91 to 1.26); 0.2740 |
| Q3 versus Q1 | 1.90 (1.67 to 2.17); <0.0001 | | | 1.70 (1.48 to 1.96); <0.0001 | | 1.24 (1.05 to 1.46); 0.0009 |

Continued

**Table 2** Continued

| | Crude estimates | Model 1, n=22 449 | Model 2, n=20 610 | Model 3, n=20 691 | Model 4, n=20 885 | Model 5, n=20 420 |
|---|---|---|---|---|---|---|
| Q4 versus Q1 | 2.95 (2.58 to 3.36); <0.0001 | | | 2.47 (2.13 to 2.86); <0.0001 | | 1.58 (1.33 to 1.88); <0.0001 |
| **Maternal occupational status** | | | | | | |
| Occupation versus no occupation | 2.15 (1.92 to 2.40); <0.0001 | | | | 1.72 (1.51 to 1.96); <0.0001 | 1.38 (1.19 to 1.61); <0.0001 |
| **Paternal occupational status** | | | | | | |
| Occupation versus no occupation | 2.24 (1.99 to 2.52); <0.0001 | | | | 1.79 (1.57 to 2.04); <0.0001 | 1.41 (1.20 to 1.65); <0.0001 |

Model 1: variables included were group (childhood obesity cohort vs comparison group), migration background, ADHD/ADD, anxiety/depression and maternal education.
Model 2: variables included were group, migration background, ADHD/ADD, anxiety/depression, and maternal and paternal education.
Model 3: variables included were group, migration background, ADHD/ADD, anxiety/depression, and maternal and paternal income.
Model 4: variables included were group, migration background, ADHD/ADD, anxiety/depression, and maternal and paternal occupational status.
Model 5: variables included were group, migration background, ADHD/ADD, anxiety/depression, maternal and paternal education, income and occupational status.
ADHD/ADD, attention deficit disorder with or without hyperactivity; SES, socioeconomic status.

However, children with obesity are often stigmatised,[41] have a low self-esteem, and are exposed to bullying and social exclusion.[42] All these factors have most likely a negative impact on school performance. In contrast, a strong social network is most probably an important factor both for good treatment response and achievement in school. An inverse association has been observed between familial social support and child weight status.[43] The relationship between BMI and school completion may also be biased from, for example, assortative mating and dynastic effects which have shown to reduce causal effects.[44] There is also a suggested interplay between genetic variants and environmental factors that may affect intelligence.[45] Thus, it is likely that both negative social effects of obesity and obesity-related morbidity, as well as genetic factors, contribute to the adverse association of childhood obesity on completed educational level.[7 29 41 45]

## Limitations

Using longitudinal data from several national registers provided an opportunity to assess the impact of obesity on completed educational level, adjusted for several important confounders. Data on both exposure and outcome were prospectively collected and defined according to standardised international classifications.[19 26]

However, some important limitations should be recognised. We did not have anthropometric data on children in the comparison group. There is no representative national data on children with obesity in Sweden. As our comparison group includes children with obesity, although likely less than 1% according to obesity diagnoses found in the National Patient Register, odds of lower level of education associated with obesity might be underestimated. It is also important to consider that children receiving obesity treatment may not be representative of all children with obesity. It should further be noted that parental SES was based on data from 1 specific year in the child's life and not over the child's entire adolescent lifetime, and that the SES indicators used may not reflect the whole SES spectrum. After providing specific ORs for each SES indicator (table 2), we can hypothesise that additive, or perhaps synergistic effects, of different SES indicators may affect the outcome. Nevertheless, the OR for the childhood obesity cohort versus comparison group remained within a narrow interval. This may indicate that there is an overlap between the measured SES domains and/or that the effect of obesity is a robust variable, independently of SES. In addition, the impact of anxiety and depression on educational level may be underestimated since these conditions often are underdiagnosed. It is furthermore plausible that controlling for anxiety and depression may lead to an underestimation of the association found between obesity in childhood and completion of 12 or more school years since the variable may act as a modifier/mediator.[7] We urge the readers to bear in mind that despite several possible mechanisms have been proposed, causal relationships of obesity and the effect of treatment on the outcome remain to be

**Table 3** Crude and adjusted ORs of completing ≥12 years of schooling in the childhood obesity cohort

| | Odds ratio (99%CI) performed with ordinary logistic regression; p value | | |
| --- | --- | --- | --- |
| | **Crude estimates** | **Model 1, n=3921** | **Model 2, n=3575** |
| Sex (girls vs boys) | 1.26 (1.06 to 1.48); 0.0004 | 1.30 (1.08 to 1.56); 0.0002 | 1.29 (1.06 to 1.56); 0.0008 |
| Migration background (non-Nordic vs Nordic) | 0.78 (0.65 to 0.94); 0.0006 | 0.74 (0.60 to 0.91); 0.0002 | 0.69 (0.55 to 0.87); <0.0001 |
| ADHD/ADD (yes vs no) | 0.31 (0.25 to 0.40); <0.0001 | 0.35 (0.27 to 0.45); <0.0001 | 0.33 (0.26 to 0.44); <0.0001 |
| Anxiety/depression (yes vs no) | 0.43 (0.35 to 0.53); <0.0001 | 0.45 (0.36 to 0.57); <0.0001 | 0.43 (0.34 to 0.55); <0.0001 |
| BMI SDS at start of treatment | 0.47 (0.38 to 0.58); <0.0001 | 0.51 (0.40 to 0.64); <0.0001 | 0.48 (0.38 to 0.62); <0.0001 |
| Age at start of treatment | 0.96 (0.93 to 1.01); 0.0234 | 1.01 (0.96 to 1.06); 0.60 | 1.01 (0.96 to 1.06); 0.805 |
| Treatment response* | | | |
| Good response versus no response | 1.46 (1.15 to 1.85); <0.0001 | 1.34 (1.04 to 1.72); 0.0031 | 1.28 (0.98 to 1.67); 0.0182 |
| Poor response versus no response | 0.78 (0.60 to 1.02); 0.0167 | 0.72 (0.54 to 0.96); 0.0028 | 0.71 (0.53 to 0.96); 0.003 |
| Dropouts versus no response | 0.71 (0.58 to 0.87); <0.0001 | 0.69 (0.55 to 0.86); <0.0001 | 0.68 (0.54 to 0.87); <0.0001 |
| Parental SES | | | |
| Medium-low versus low parental SES | 1.63 (1.31 to 2.03); <0.0001 | 1.56 (1.24 to 1.98); <0.0001 | |
| Medium-high versus low parental SES | 2.40 (1.90 to 3.05); <0.0001 | 2.19 (1.69 to 2.83); <0.0001 | |
| High versus low parental SES | 2.99 (2.11 to 4.22); <0.0001 | 2.83 (1.95 to 4.12); <0.0001 | |
| Maternal education | | | |
| Upper secondary school versus compulsory school | 1.74 (1.44 to 2.09); <0.0001 | | 1.57 (1.27 to 1.94); <0.0001 |
| University degree versus compulsory school | 2.47 (1.94 to 3.16); <0.0001 | | 2.23 (1.67 to 2.98); <0.0001 |
| Paternal education | | | |
| Upper secondary school versus compulsory school | 1.48 (1.23 to 1.79); <0.0001 | | 1.26 (1.03 to 1.55); 0.004 |
| University degree versus compulsory school | 1.92 (1.44 to 2.56); <0.0001 | | 1.41 (1.01 to 1.95); 0.0076 |

Model 1: variables included were sex, migration background, ADHD/ADD, anxiety/depression, BMI SDS and age at start of treatment, treatment response and parental SES.
Model 2: variables included were sex, migration background, ADHD/ADD, anxiety/depression, BMI SDS and age at start of treatment, treatment response, and maternal and paternal education.
*Good response=decrease of BMI SDS ≥0.25 units; no response=BMI SDS ±0.24 units; poor response=increase of BMI SDS ≥0.25 units; dropouts=no follow-up measure or less than 1 year between their first and last measure.
ADHD/ADD, attention deficit disorder with or without hyperactivity; BMI SDS, body mass index SD score; SES, socioeconomic status.

established. Lastly, factors such as free education, school lunches and students' healthcare may have an impact on the generalisability of our data to other populations.

### Conclusion

In this longitudinal, population-based study, individuals

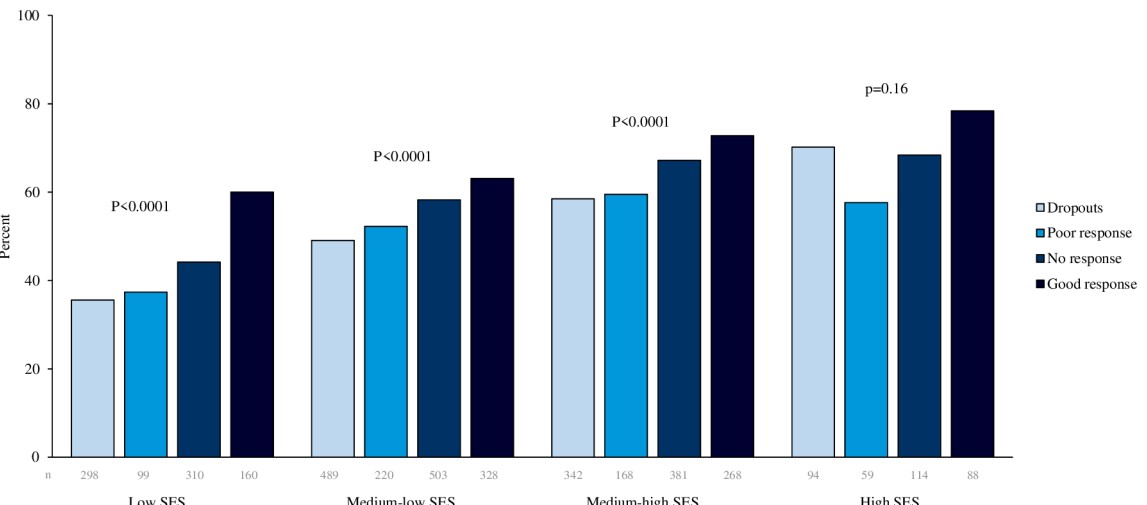

**Figure 3** Crude percentage of individuals with obesity in childhood (n=3921) completing ≥12 school years by parental SES and treatment response. P=p for trend; n refers to number of individuals in each category. SES, socioeconomic status.

with obesity in childhood were less likely to complete 12 or more years of schooling compared with a group from the general population. The odds associated with obesity remained significantly increased even after taking parental SES and other important risk factors, such as ADHD, into consideration. The underlying mechanisms are unclear but previous studies indicate that the effects of obesity on school completion both involve psychosocial effects and cognitive functions.[16 17 32] The negative impact of childhood obesity on educational level may partly be reversed by successful obesity treatment in childhood. Results from this study underline effects of childhood obesity on schooling. For the sake of an optimal educational environment, it is of great importance to increase awareness both in schools and among decision makers to allocate resources for potential extra support, for example, reduce stigma and increase educational support.

**Acknowledgements** The authors would like to thank all local healthcare professionals in Sweden working with children and adolescents with obesity and their valuable work with BORIS, and the BORIS steering committee for establishing and maintaining the register.

**Contributors** Conceptualisation—LL, PD, MP, CM and EH. Data curation—LL. Formal analysis—LL and EH. Funding acquisition—LL, CM and EH. Methodology—LL and EH. Project administration—LL and EH. Supervision—MP, EH, PD and CM. Writing (original draft)—LL. Writing (review and editing)—LL, PD, MP, CM and EH.

**Funding** This study was supported by funds to LL by Crown Princess Lovisa's Foundation (2017-00348), Samariten Foundation (2017-0305), the Stockholm FreeMason Foundation for Children's Welfare, Sällskapet Barnavård, Anna-Lisa och Arne Gustafssons Foundation, Solstickan Foundation and Sven Jerring Foundation. EH was supported by the Swedish Society for Medical Research, Fredrik and Ingrid Thuring's Foundation (2017-00309) and Magnus Bergvall Foundation (2017-02113); and CM by the Swedish Heart and Lung Foundation (20150790).

**Competing interests** None declared.

**Patient consent for publication** Not required.

**Ethics approval** The study was approved by the regional Ethics Committee in Stockholm, Sweden (No. 2016/922-31/1).

**Provenance and peer review** Not commissioned; externally peer reviewed.

**Data availability statement** Data may be obtained from a third party and are not publicly available. The data that support the findings of this study contain sensitive information. Restrictions therefore apply to the availability of these data, which were used under licence for the current study, and so are not publicly available. According to Swedish law and the General Data Protection Regulation, the authors are not permitted to share the datasets used in this study with third parties. Given that an ethical approval is obtained, any individual may apply for data from Statistics Sweden via information@scb.se, the Swedish National Board of Health and Welfare via registerservice@socialstyrelsen.se, and the Swedish Childhood Obesity Treatment Register via http://www.e-boris.se/kontaktuppgifter/.

**ORCID iD**
Emilia Hagman http://orcid.org/0000-0003-1433-2295

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
