## [Reviewer comments · BMJ Open]

ARTICLE DETAILS

TITLE (PROVISIONAL)	Obesity in childhood, socioeconomic status, and completion of 12 or more school years: a prospective cohort study
AUTHORS	Lindberg, Louise; Persson, Martina; Danielsson, Pernilla; Hagman, Emilia; Marcus, Claude

VERSION 1 – REVIEW

REVIEWER	Rønnaug Ødegård Norwegian University of Science and Technology, NTNU, Norway
REVIEW RETURNED	12-Sep-2020

GENERAL COMMENTS	Review comments In a large prospective cohort study the authors show reduced odds of completing upper secondary school for treatment seeking children with obesity (included in a obesity registry) compared to population controls, and the effect was not explained by parental Socio-economic status (SES). Compared to those with no treatment response, there were indications that good response to treatment was associated with higher odds of school completion. Although the authors attempt to address some important and timely questions of the field, the manuscript contains some issues that should be addressed. Title Line 1-3: «educational level» is unprecise as the phrase often refers to final education level. Please use the specific study outcome «upper secondary school». Abstract Line 45-46: Word left out: However, obesity in childhood remains a strong risk factor of not completing ≥ 12 school years, independently of parental SES, aOR=0.57 [0.51 to 0.63]. Introduction Line 68: "Childhood obesity is associated with increased risks of somatic morbidity and risk of premature death in adulthood.2-5" This is not necessarily true. Richardson et al. (1) do not find a causal association between childhood obesity and CVD and DM2 later in life using a new polygenic risk score for childhood BMI (obesity) in a mendelian randomization study. The effects of childhood obesity on certain adult diseases are likely mediated through obesity in adulthood. Line 73 and 74: «We have previously confirmed the association between obesity and lower attained level of education among both girls and boys» The causal association between lower BMI and increased educational attainment (or vice versa) has also been shown in
--

	mendelian randomization studies. However, a recent study from the Norwegian HUNT population show that these causal effects were greatly reduced within family analyses (2). Please consider bias from assortative mating, dynastic effects and population stratification. Line 79: "In particular, parental education has been demonstrated to influence the child's performance at school.13-15" Plomin (3) argues the importance of not only the family environment but also the importance of genetic effects on school performance. Methods Line 96-97: "Treatment is individualized and consists primarily of lifestyle modification (i.e. diet and physical activity)" Treatment-seeking children with obesity frequently present problems related to school participation, incl. bullying and school subject achievements. Multidisciplinary obesity treatment therefore often includes collaboration with schools. Are the authors sure that the outcome (graduation from upper secondary school) is independent of provided treatment? Please clarify and discuss. Line 111-112 (discussed in limitations line 352):" However, less than 1% of the individuals in the comparison group were found in the National Patient Register with a diagnosis of obesity in childhood" This is most likely an underestimation of the prevalence of obesity in the control group; national average estimates would be more appropriate as long as the control group is considered a population control group. Line 133: " ≥12 years of schooling" Is the main outcome graduation from of upper secondary school by the age of 19, or 20? Please clarify main outcome. Line 140: "Definition of main exposures" Please describe the main exposure more precisely in this paragraph; e.g. being a treatment-seeking child with obesity or being included in the childhood obesity registry. Also the definition of obesity (inclusion criteria of the obesity registry) should be more clear here. Line 149-172: The authors create a composite parental SES score that encompasses three aspects of SES, parental education, income and occupational status. The new SES score is more strongly associated to completion of upper secondary school than each of the three components separately as shown in Table 2. The effect estimates of being in the obesity cohort compared to controls on the odds of school completion did however not differ between the different SES indicators (OR 0,57-0,54 in Model 1, 2 and 3 respectively (Table 2)) and the added value of the composite SES variable is not clearly presented. Please discuss. Results Line 221 and 222: "a greater proportion of children with obesity grew up in households with low SES compared with the comparison group (22.0% vs. 14.4%; p<0.0001)" Could this difference between the groups introduce confounding that was not controlled for in the main analyses?
--	---

Line 235-239: “The adjusted odds ratio (aOR) [99% CI] to complete ≥ 12 years of schooling for the childhood obesity cohort versus the comparison group were lower in the higher level of SES: aOR low parental SES = 0.69 [0.50 to 0.95], $p=0.0026$; aOR medium-low parental SES = 0.59 [0.48 to 0.72], $p<0.0001$; aOR medium-high parental SES = 0.46 [0.35 to 0.60], $p<0.0001$; aOR high parental SES = 0.27 [0.14 to 0.54], $p<0.0001$ ”

Are these stratified analyses according to parental SES? Please describe more clearly. Given the slightly overlapping confidence intervals between the three analyses please describe the results less absolute.

Line 241-245: Did you find the same result when applying the other SES indicators?

Line 280: Table 3 is better to read if Treatment response is described first (analogue to table 2)

Discussion

Line 323-324: “Possible mechanisms may include psychosocial aspects such as

stigma³⁰ and increased risk for anxiety and depression”
Education is a social construct. A discussion of what you are actually measuring with educational attainment (ie. resourcefulness, intelligence, ability to conform) would be informative for the subsequent discussion on mechanisms.

Line 344 – 345: “Thus, it is likely that both negative social effects of obesity and obesity-related morbidity, as well as genetic factors, contribute to the adverse association of childhood obesity on completed educational level”
References? A plausibility that there is an overlap in the genetic variants associated with childhood obesity and educational attainment should be mentioned (3).

Line 357-358: “It is also important to consider that children receiving obesity treatment may not be representative of all children with obesity. However, such bias is reduced as a large proportion of children are referred to treatment from school»
I assume they are referred from school nurses, but it is not clear how this will reduce bias. The sentence could be left out to underline the important message in the sentence above

Line 369 - 370 “The underlying mechanisms are unclear but previous studies indicate that the effects of obesity both involve psychosocial effects and cognitive functions. References?”

Line 372-375 “Results from this study underline the wide effects of childhood obesity on public health and the importance of continued efforts to reduce the prevalence of obesity in children”
This is beyond the scope of the present study. Try to clarify and specify your final message and how it relates to your findings.

References

- 1) Richardson, T.G. et al. (2020) Use of genetic variation to separate the effects of early and later life adiposity on disease risk: mendelian randomisation study. *BMJ*, 369, m1203.
- 2) Brumpton, B et al (2020) Avoiding dynastic, assortative mating, and population stratification biases in Mendelian randomization through within-family analyses *Nat Commun.* Jul 14;11(1):3519.

	3) Plomin, R et al (2018) The new genetics of intelligence. Nature reviews. Genetics, 19, 148-159.
--	--

REVIEWER	Anders Hjern Karolinska institutet, Sweden
REVIEW RETURNED	28-Sep-2020

GENERAL COMMENTS	This is an interesting study on a very important public health topic for children. It is a well-written and clear article, but the regression modelling is not well-designed and the interpretation of one of the main results is problematic. Comments.  1. The study uses a rich set of socioeconomic indicators from national registers to adjust for confounding. To use several indicators instead of one is a strength, but the ones chosen are not the ideal ones. Disposable incomes of both the mother and the father are used here, but a summarized disposable household income is to prefer, since a) these incomes do not take into account the household size and 2. a very high paternal income is often coupled with a comparative low maternal income because of a choice of short working hours. This household disposable income is available from the same data source as the income variables used here. 2. There is no indicator for family situation in the analysis. A recent systematic review (Duriancik&Goff, 2019) has shown that children in single parent households more often develop childhood obesity and children in single parent households in Sweden have also been shown to have a lower educational achievement compared with children in two parent households(Weitof et al 2004). It could be added that the sociodemographic register data source in this study includes a family status indicator (Famstf). 3. Ethnicity/Nationality has been shown to be quite powerful determinant of childhood obesity. This is shown in the large variation in obesity rates between different high income countries (see HBSC studies) and in differences between immigrant groups and the majority population in Sweden (see for instance Khanolkar et al 2013). Some ethnic minorities have a higher risk, in the study by Khanolkar et al, most non-European heritages were associated with a higher risk while a Finnish background as associated with a lower risk. It is likely that a mixture of lifestyle and genetic factors are behind these variations by ethnicity. In this study, the population is divided into Nordic and Non-Nordic background. This seems very arbitrary and has little to do with the variation of obesity by ethnicity in the Swedish population. Even the Nordic category is problematic, considering the lower risk of obesity in them in the study by Khanolkar. This is a major challenge in this study, since 25% of the population have a non-Nordic background and thus even more a non-Swedish background. There are two ways forward with this problem. A more appropriate categorization can be used, based on the variation within the study population and the literature. And/or a sensitivity analysis could be made, including only children with a Swedish background. I think doing both would be the most convincing solution. 4. The study uses register data on psychiatric disorders and psychotropic medication to adjust for mental health problems. More specifically, the study adjust for ADHD and anxiety and depression. Adjusting for ADHD seems to be a good idea, since ADHD probably is a cause rather than a consequence of both
---

	obesity and school failure. And furthermore, register data probably picks up the large majority of children with ADHD in Sweden today with the current high rates of medication in school age. Adjusting for anxiety and depression is more problematic. First of all, it seems likely that the overrepresentation of these disorders in obese children is a consequence rather than a cause of obesity, indicating that they are in the pathway from obesity to school failure. Hence, adjusting for them leads to overadjustment, Secondly, the prevalence rates of these conditions in the population than in the registers, suggesting that they are not very effective indicators. 5. The study finds an association between successful obesity treatment and better educational achievement. This leads to a conclusion that successful treatment of obesity may lead to improved school performance. This is indeed a possibility. But a more probable explanation for this finding is that the same factors that promote learning and school work also facilitate the behavior change involved with obesity treatment. Indeed, Kark et al (2013), in a study in Stockholm, has shown that a poor school performance leads to increased weight gain during puberty. These shared factors here could be factors within the child, such as cognitive competence and personality traits but also unmeasured factors in the environment, such as relations within the family and the peer group. The only way to show that obesity treatment can improve school performance is to do a randomized controlled trial. This needs to be clearly pointed out in the limitations section and the conclusions should leave no doubt about this alternative interpretation.
--	---

VERSION 1 – AUTHOR RESPONSE

Reviewer: 1

Reviewer Name: Rønnaug Ødegård

Institution and Country: Norwegian University of Science and Technology, NTNU, Norway Please state any competing interests or state 'None declared': None declared

Please leave your comments for the authors below Review comments In a large prospective cohort study the authors show reduced odds of completing upper secondary school for treatment seeking children with obesity (included in a obesity registry) compared to population controls, and the effect was not explained by parental Socio-economic status (SES). Compared to those with no treatment response, there were indications that good response to treatment was associated with higher odds of school completion. Although the authors attempt to address some important and timely questions of the field, the manuscript contains some issues that should be addressed.

Title

Line 1-3: «educational level» is unprecise as the phrase often refers to final education level. Please use the specific study outcome «upper secondary school».

Answer: Thank you for this reflection. The title now reads as follow: Obesity in childhood, socioeconomic status, and completion of 12 or more school years: a prospective cohort study

Abstract

Line 45-46: Word left out: However, obesity in childhood remains a strong risk factor of not completing ≥ 12 school years, independently of parental SES, aOR=0.57 [0.51 to 0.63].

Answer: Thank you for pointing this out. The sentence is now revised.

Introduction

Line 68: "Childhood obesity is associated with increased risks of somatic morbidity and risk of premature death in adulthood.2-5" This is not necessarily true. Richardson et al. (1) do not find a causal association between childhood obesity and CVD and DM2 later in life using a new polygenic risk score for childhood BMI (obesity) in a mendelian randomization study. The effects of childhood obesity on certain adult diseases are likely mediated through obesity in adulthood.

Answer: Thank you for this complimentary data. The cited references refer to epidemiological studies and one review. Just as the present study, epidemiological associations are not always supported with mechanistic explanatory pathways. It is not our intention to imply that there is a causal relationship, but rather an association. Hence, we wish to keep the sentence as it is, but are willing to change it if it still feels misleading.

Line 73 and 74: «We have previously confirmed the association between obesity and lower attained level of education among both girls and boys» The causal association between lower BMI and increased educational attainment (or vice versa) has also been shown in mendelian randomization studies. However, a recent study from the Norwegian HUNT population show that these causal effects were greatly reduced within family analyses (2). Please consider bias from assortative mating, dynastic effects and population stratification.

Answer: Thank you for the comment. We have added a sentence about this in the discussion, line 348-350. "The relationship between BMI and school completion may also be biased from e.g. assortative mating and dynastic effects which have shown to reduce causal effects".

Line 79: "In particular, parental education has been demonstrated to influence the child's performance at school.13-15" Plomin (3) argues the importance of not only the family environment but also the importance of genetic effects on school performance.

Answer: Thank you for the suggestion. Genetic effects on school performance are indeed important to highlight which is briefly mentioned in the discussion line 352. In addition to this, we have added a sentence (line 350-351) on the potential overlap in genetic variants associated with obesity and education as the reviewer suggested in a comment below. We decided not to discuss genetic effects in the background since it is somewhat out of the main scope of this paper. However, we are willing to develop the argument in more detail in the background and/or discussion if the reviewer and editor prefer so.

Methods

Line 96-97: "Treatment is individualized and consists primarily of lifestyle modification (i.e. diet and physical activity)". Treatment-seeking children with obesity frequently present problems related to school participation, incl. bullying and school subject achievements. Multidisciplinary obesity treatment therefor often includes collaboration with schools. Are the authors sure that the outcome (graduation from upper secondary school) is independent of provided treatment? Please clarify and discuss.

Answer: Thank you for the question. This is a delicate question. Since cohort studies cannot prove causation, we cannot state that the outcome is independent of the provided treatment. In the present study, we were not able to evaluate type of treatment the children had received (i.e. individual, group, multidisciplinary teams etc.) but it would be very interesting to examine in future studies. Our results did however show that individuals who had a successful response to treatment were more likely to complete school compared to individuals with no response to treatment. Nevertheless, there are still other factors that may impact the association between treatment response and school completion including family structure, sleep, genetic components, and cognitive abilities.

Line 111-112 (discussed in limitations line 352): "However, less than 1% of the individuals in the comparison group were found in the National Patient Register with a diagnosis of obesity in childhood" This is most likely an underestimation of the prevalence of obesity in the control group; national average estimates would be more appropriate as long as the control group is considered a population control group.

Answer: We agree that this number probably is an underestimation. Regarding obesity, the National Patient Register is not an optimal source for estimating the prevalence, since all individuals with obesity does not seek help from the health care. Therefore, the number <1% is not to describe the prevalence, but to give an indication how many of the individuals in the comparison group that has been found to have obesity, but not received obesity treatment

according to the BORIS register. Hence the prevalence of obesity in the comparison group remains unknown but may be similar as the general pediatric population. One may also hypothesize that the prevalence of obesity in the comparison group may be less than in the general pediatric population, since the pool of possible controls have been drained from children and adolescents with obesity (which are found in the BORIS-register). And since these individuals may not be controls, the proportion of subjects with obesity may be lower in the comparison group.

Nevertheless, the proportion of individuals that complete ≥ 12 years in school in the comparison group corresponds to the general statistics provided by Statistics Sweden (<https://www.skolverket.se/om-oss/press/pressmeddelanden/pressmeddelanden/2018-12-12-fortsatt-trend--fler-elever-tar-gymnasieexamen>). Hence, from the perspective of achieving education, the comparison group seems to be representative, while from an obesity prevalence point of view, we cannot know for sure.

Line 133: “ ≥ 12 years of schooling”

Is the main outcome graduation from of upper secondary school by the age of 19, or 20? Please clarify main outcome.

Answer: The main outcome is graduation from upper secondary school (US High School), which in Sweden is a total of 12 completed school years. The students graduating from upper secondary school is usually 18 or 19 years of age. This is stated on line 135. However, we noticed the word “upper” was missing in that sentence which has now been corrected.

Line 140: “Definition of main exposures” Please describe the main exposure more precisely in this paragraph; e.g. being a treatment-seeking child with obesity or being included in the childhood obesity registry. Also the definition of obesity (inclusion criteria of the obesity registry) should be more clear here.

Answer: Thank you for the comment. The study population and the obesity register are described in the first paragraph of the Methods and in Figure 1. Nevertheless, we clarified the heading which now reads as “Definition of exposure variables”.

Line 149-172: The authors create a composite parental SES score that encompasses three aspects of SES, parental education, income and occupational status. The new SES score is more strongly associated to completion of upper secondary school than each of the three components separately as shown in Table 2. The effect estimates of being in the obesity cohort compared to controls on the odds of school completion did however not differ between the different SES indicators (OR 0,57-0,54 in Model 1, 2 and 3 respectively (Table 2)) and the added value of the composite SES variable is not clearly presented. Please discuss.

Answer: Thank you for this reflection. We use a rich set of socioeconomic indicators instead of one, which we believe is a strength. SES is complex and it is plausible that certain domains of SES are more or less associated with a specific outcome. After providing specific ORs for each SES domain we can hypothesize that additive, or perhaps synergistic effects, of different SES domains may affect the outcome. Nevertheless, the OR for the childhood obesity cohort vs. comparison group remained within a narrow interval. This may indicate that there is an overlap between the measured SES domains and/or that the effect of obesity is a robust variable, independently of SES. If the referee or editor would like us to add this analysis of Table 2 in the discussion, we will be happy to do so.

Results

Line 221 and 222: “a greater proportion of children with obesity grew up in households with low SES compared with the comparison group (22.0% vs. 14.4%; $p < 0.0001$)” Could this difference between the groups introduce confounding that was not controlled for in the main analyses?

Answer: Certainly. There may always be confounders that are not controlled for. Collinearity often lead to wide confidence intervals, that results in less reliable probabilities in terms of the effect of independent variables in a model. The main analyze was adjusted for SES and the adjusted OR is highly significant and have rather narrow confidence intervals, we feel confident in that we have provided the most suitable statistics given the data we have.

Line 235-239: “The adjusted odds ratio (aOR) [99% CI] to complete ≥ 12 years of schooling for the childhood obesity cohort versus the comparison group were lower in the higher level of SES: aOR low parental SES = 0.69 [0.50 to 0.95], $p = 0.0026$; aOR medium-low parental SES = 0.59 [0.48 to 0.72],

p<0.0001; aOR medium-high parental SES = 0.46 [0.35 to 0.60], p<0.0001; aOR high parental SES = 0.27 [0.14 to 0.54], p<0.0001” Are these stratified analyses according to parental SES? Please describe more clearly. Given the slightly overlapping confidence intervals between the three analyses please describe the results less absolute.

Answer: Yes, that is correct, these analyses are stratified by parental SES. The sentence has been revised and now state: “The adjusted odds ratio (aOR) [99% CI] to complete ≥12 years of schooling stratified by parental SES, when comparing the childhood obesity cohort with the comparison group, showed a trend towards lower OR in the higher level of SES.” We hope this sentence is now clearer and that the results are described in less absolute terms.

1. Line 241-245: Did you find the same result when applying the other SES indicators?

Answer: Thank you for the question. Overall, the results were similar when applying the other SES indicators. In detail, education and occupation followed the same trend as the composite SES variable. The trend was clearer when stratified by the components of fathers’ education and occupation than the mothers’ education and occupation. For the income indicator, the OR was lower for the top and bottom quartile of income than it was for the middle quartiles (25-50% and 50-75%). If the referee or editor think this is valuable results to have as supplementary data, we are happy to add that.

2. Line 280: Table 3 is better to read if Treatment response is described first (analogue to table 2)

Answer: Thank you for the suggestion. We have added a footnote in table 3 defining treatment response. The footnote reads as follows: Good response=decrease of BMI SDS ≥ 0.25 units; No response=BMI SDS +/- 0.24 units; Poor response=increase of BMI SDS ≥ 0.25 units; Dropouts=no follow-up measure or less than one year between their first and last measure.

Discussion

Line 323-324: “Possible mechanisms may include psychosocial aspects such as stigma³⁰ and increased risk for anxiety and depression” Education is a social construct. A discussion of what you are actually measuring with educational attainment (ie. resourcefulness, intelligence, ability to conform) would be informative for the subsequent discussion on mechanisms.

Answer: Thank you for this excellent suggestion. We have now added the following sentence to the discussion line 325-326. “Several parameters may interplay when measuring school completion, e.g. resourcefulness, intelligence, or ability to conform”.

Line 344 – 345: “Thus, it is likely that both negative social effects of obesity and obesity-related morbidity, as well as genetic factors, contribute to the adverse association of childhood obesity on completed educational level” References? A plausibility that there is an overlap in the genetic variants associated with childhood obesity and educational attainment should be mentioned (3).

Answer: References have now been added on line 353. In addition, we have added a sentence on genetic variants intelligence based on Plomin et al., 2018 on line 350-351.

Line 357-358: “It is also important to consider that children receiving obesity treatment may not be representative of all children with obesity. However, such bias is reduced as a large proportion of children are referred to treatment from school» I assume they are referred from school nurses, but it is not clear how this will reduce bias. The sentence could be left out to underline the important message in the sentence above

Answer: Thank you for this suggestion. The last sentence is now removed.

Line 369 - 370 “The underlying mechanisms are unclear but previous studies indicate that the effects of obesity both involve psychosocial effects and cognitive functions. References?

Answer: Thank you for pointing this out. References have now been added to the statement.

Line 372-375 “Results from this study underline the wide effects of childhood obesity on public health and the importance of continued efforts to reduce the prevalence of obesity in children” This is beyond the scope of the present study. Try to clarify and specify your final message and how it relates to your findings.

Answer: Thank you for this comment. We have now revised the final part of the conclusion, which now states: Results from this study underline effects of childhood obesity on schooling.

For the sake of an optimal educational environment, it is of great importance to increase awareness both in schools and among decision makers to allocate resources for potential extra support, e.g. reduce stigma and increase educational support. Please see lines 386-390.

References

- 1) Richardson, T.G. et al. (2020) Use of genetic variation to separate the effects of early and later life adiposity on disease risk: mendelian randomisation study. *BMJ*, 369, m1203.
- 2) Brumpton, B et al (2020) Avoiding dynastic, assortative mating, and population stratification biases in Mendelian randomization through within-family analyses *Nat Commun.* Jul 14;11(1):3519.
- 3) Plomin, R et al (2018) The new genetics of intelligence. *Nature reviews. Genetics*, 19, 148-159.

Reviewer: 2

Reviewer Name: Anders Hjern

Institution and Country: Karolinska institutet, Sweden Please state any competing interests or state 'None declared': None declared

Please leave your comments for the authors below This is an interesting study on a very important public health topic for children. It is a well-written and clear article, but the regression modelling is not well-designed and the interpretation of one of the main results is problematic.

Comments.

1. The study uses a rich set of socioeconomic indicators from national registers to adjust for confounding. To use several indicators instead of one is a strength, but the ones chosen are not the ideal ones. Disposable incomes of both the mother and the father are used here, but a summarized disposable household income is to prefer, since a) these incomes do not take into account the household size and 2. a very high paternal income is often coupled with a comparative low maternal income because of a choice of short working hours. This household disposable income is available from the same data source as the income variables used here.

Answer: Thank you for these very delicate questions. Using the variable (DisplnkKE04) which take household size into account would have been a good option. However, there were several reasons why we choose to use the LISA variable Displnk04 instead of DisplnkKE04.

First, we wanted to examine if there were any differences with regards to the effect of mother's SES and father's SES, respectively, on the association between childhood obesity and school completion. Thus, using a disposable household income would not have enabled this comparison in the way we wanted which was to mirror the individual disposable income the mother and the father de facto had.

Second, a variable reflecting a situation in the household assumes everything is split equal. However, households may have a divided economy.

Third, viewing previous published research using the same register, there was roughly an equal number of articles who had used DisplnkKE04, DisplnkPers04, Displnk04 and DisplnkFam04.

Fourth, for example, if there are children over 18 years of age in the household who earn money but do not share his or her income to the household income, may result in a falsely high disposable income. Further on, there might be other adults living in the same household but who are not a part of the family and thus do not contribute with his or her salary.

With these arguments we would like to keep estimating income with the variable Displnk04, but we do agree that there are both limitations and strengths with all the available income variables in LISA.

2. There is no indicator for family situation in the analysis. A recent systematic review (Duriancik&Goff, 2019) has shown that children in single parent households more often develop childhood obesity and children in single parent households in Sweden have also been shown to have a lower educational achievement compared with children in two parent households(Weitof et al 2004). It could be added that the sociodemographic register data source in this study includes a family status indicator (Famstf).

Answer: Thank you for this suggestion. We are familiar with the "FamStF"-indicator, which is a derived variable for other sources and comprises 28 levels in total. Even if it is possible to pool some of these levels, we made an active decision to not include this variable, which of course may be discussed. The reasons for this decision include 1) It is not possible to

distinguish between e.g. a single mom (without a partner) and a mom living with a partner without common children. 2) It may state that two adults live together and have common children. Nevertheless, if that is the case for both parents, we do not know if the two adults are the parents of the child or if both have new families. 3) Family status is dynamic. With these arguments and the fact that family status may be quite complex in Sweden nowadays since the alternatives to a nuclear family are endless, we decided to not include this variable.

3. Ethnicity/Nationality has been shown to be quite powerful determinant of childhood obesity. This is shown in the large variation in obesity rates between different high income countries (see HBSC studies) and in differences between immigrant groups and the majority population in Sweden (see for instance Khanolkar et al 2013). Some ethnic minorities have a higher risk, in the study by Khanolkar et al, most non-European heritages were associated with a higher risk while a Finnish background was associated with a lower risk. It is likely that a mixture of lifestyle and genetic factors are behind these variations by ethnicity. In this study, the population is divided into Nordic and Non-Nordic background. This seems very arbitrary and has little to do with the variation of obesity by ethnicity in the Swedish population. Even the Nordic category is problematic, considering the lower risk of obesity in them in the study by Khanolkar. This is a major challenge in this study, since 25% of the population have a non-Nordic background and thus even more a non-Swedish background. There are two ways forward with this problem. A more appropriate categorization can be used, based on the variation within the study population and the literature. And/or a sensitivity analysis could be made, including only children with a Swedish background. I think doing both would be the most convincing solution.

Answer: Thank you for this excellent suggestion. We agree with the reviewer that categorizing ethnicity into only two groups could be arbitrary. However, the reason for this decision was due to that the Nordic countries are somewhat similar (and hence comparable) when it comes to culture and language compared to many other countries in Europe and the rest of the world. Nevertheless, we have followed the reviewer's suggestions and performed sensitivity analyses excluding children who did not have a Swedish background (defined as child born in Sweden with at least one parent also born in Sweden).

Results persisted in the sensitivity analyses. A sentence about this is now added to the manuscript. "Further, excluding individuals with a non-Swedish background...did not alter the association between childhood obesity and attained level of education. For example, the adjusted odds ratio (aOR) [99% CI] to complete ≥ 12 years in school for the obesity cohort vs. the comparison group was 0.55 [0.48 to 0.63], $p < 0.0001$." Please see line 255-260.

4. The study uses register data on psychiatric disorders and psychotropic medication to adjust for mental health problems. More specifically, the study adjusts for ADHD and anxiety and depression. Adjusting for ADHD seems to be a good idea, since ADHD probably is a cause rather than a consequence of both obesity and school failure. And furthermore, register data probably picks up the large majority of children with ADHD in Sweden today with the current high rates of medication in school age. Adjusting for anxiety and depression is more problematic. First of all, it seems likely that the overrepresentation of these disorders in obese children is a consequence rather than a cause of obesity, indicating that they are in the pathway from obesity to school failure. Hence, adjusting for them leads to overadjustment. Secondly, the prevalence rates of these conditions in the population than in the registers, suggesting that they are not very effective indicators.

Answer: Thank you for this reflection. We chose to adjust the analyses for depression since it is known that there is an association between anxiety and depression. The association is suggested to be bi-directional, thus, depression may be a consequence of obesity, but depression may also cause obesity. We looked at anxiety and depression from the age of 13 to 19.9 years. The reason was to narrow down the time from exposure to outcome and that 13 was a good age since this is when children start the seventh grade (which usually means changing school).

Regarding the reviewer's last comment, it seems there are some words missing but we will try to answer the question as we believe it was meant. The worldwide prevalence of anxiety and depression in children, regardless of weight status, is estimated to stand at 6.5% and 2.6% (Polanczyk et al., 2015). We have not found any worldwide prevalence numbers of anxiety or depression in children with obesity. If we pool the reported prevalence of anxiety and depression by Polanczyk and colleagues, it sums up to 9.1%. The percentage of anxiety/depression in the comparison group in our study was 11.5%. Hence, close but slightly

higher, which could depend on several factors including differences health care systems, reporting and ways to assess the diagnosis.

5. The study finds an association between successful obesity treatment and better educational achievement. This leads to a conclusion that successful treatment of obesity may lead to improved school performance. This is indeed a possibility. But a more probable explanation for this finding is that the same factors that promote learning and school work also facilitate the behavior change involved with obesity treatment. Indeed, Kark et al (2013), in a study in Stockholm, has shown that a poor school performance leads to increased weight gain during puberty. These shared factors here could be factors within the child, such as cognitive competence and personality traits but also unmeasured factors in the environment, such as relations within the family and the peer group. The only way to show that obesity treatment can improve school performance is to do a randomized controlled trial. This needs to be clearly pointed out in the limitations section and the conclusions should leave no doubt about this alternative interpretation.

Answer: Thank you for this comment. We agree that a prospective cohort study, like the one we have performed, cannot provide causality like an RCT can. However, we still believe the findings are both relevant and important, both for the patient group and for policy makers to improve treatment of obesity in children. We have clarified this further in the limitation as the reviewer suggested. Please see line 371-373. “We urge the reader to bear in mind that despite several possible mechanisms have been proposed, causal relationships of obesity and the effect of treatment on the outcome remains to be established.”. Further, in the conclusion, it now states that obesity treatment may affect the outcome. Please see line 382.

VERSION 2 – REVIEW

REVIEWER	Rønnaug Ødegård Norwegian University of Technology and Science, NTNU Norway
REVIEW RETURNED	19-Nov-2020

GENERAL COMMENTS	Dear Author Thank you for your thorough response to my comments. The paper is now more nuanced and precise. I have no further questions, but two suggestions for a publication:  I support your suggestion (see below) to add the SES indicator analyses as supplementary data 1. Line 241-245: Did you find the same result when applying the other SES indicators? Answer: Thank you for the question. Overall, the results were similar when applying the other SES indicators. In detail, education and occupation followed the same trend as the composite SES variable. The trend was clearer when stratified by the components of fathers' education and occupation than the mothers' education and occupation. For the income indicator, the OR was lower for the top and bottom quartile of income than it was for the middle quartiles (25-50% and 50-75%). If the referee or editor think this is valuable results to have as supplementary data, we are happy to add that.  I think the discussion will benefit from adding some of the discussion of the effect sizes of the various SES scores below Line 149-172: The authors create a composite parental SES score that encompasses three aspects of SES, parental education, income and occupational status. The new SES score is more strongly associated to completion of upper secondary school than each of the three components separately as shown in Table 2. The effect estimates of being in the obesity cohort compared to controls on the odds of school completion did however not differ
--

	between the different SES indicators (OR 0,57-0,54 in Model 1, 2 and 3 respectively (Table 2)) and the added value of the composite SES variable is not clearly presented. Please discuss. Answer: Thank you for this reflection. We use a rich set of socioeconomic indicators instead of one, which we believe is a strength. SES is complex and it is plausible that certain domains of SES are more or less associated with a specific outcome. After providing specific ORs for each SES domain we can hypothesize that additive, or perhaps synergistic effects, of different SES domains may affect the outcome. Nevertheless, the OR for the childhood obesity cohort vs. comparison group remained within a narrow interval. This may indicate that there is an overlap between the measured SES domains and/or that the effect of obesity is a robust variable, independently of SES. If the referee or editor would like us to add this analysis of Table 2 in the discussion, we will be happy to do so.
--	--

REVIEWER	Anders Hjern Karolinska institutet, Sweden
REVIEW RETURNED	12-Nov-2020

GENERAL COMMENTS	The authors have addressed most of my previous concerns adequately. However, one concern remains. 1. The authors have included anxiety/depression as a confounder in the regression analysis, which is problematic since it is probably more of a mediator/modifier. At least, that's what the authors have suggested recently in a recent article (Lindberg et al BMC Med 2020). This can be expected to lead to an underestimation of the association between obesity and education in this study. Either the data should be reanalysed without this variable or discussed in the limitations section.
--

VERSION 2 – AUTHOR RESPONSE

Reviewer: 1

Reviewer Name: Rønnaug Ødegård

Institution and Country: Norwegian University of Technology and Science, NTNU Norway Competing

interests 1: None declared Comments to the Author See enclosed letter (pasted in below)

Dear Author

Thank you for your thorough response to my comments. The paper is now more nuanced and precise. I have no further questions, but two suggestions for a publication:

- I support your suggestion (see below) to add the SES indicator analyses as supplementary data

1. Line 241-245: Did you find the same result when applying the other SES indicators? Answer:

Thank

you for the question. Overall, the results were similar when applying the other SES indicators. In detail, education and occupation followed the same trend as the composite SES variable. The trend was clearer when stratified by the components of fathers' education and occupation than the mothers' education and occupation. For the income indicator, the OR was lower for the top and bottom quartile of income than it was for the middle quartiles (25-50% and 50-75%). If the referee or editor think this is valuable results to have as supplementary data, we are happy to add that.

Answer: Thank you for the suggestion. This has been added as supplementary data. Please see S3 Table.

- I think the discussion will benefit from adding some of the discussion of the effect sizes of the various SES scores below

Line 149-172: The authors create a composite parental SES score that encompasses three aspects of

SES, parental education, income and occupational status. The new SES score is more strongly associated to completion of upper secondary school than each of the three components separately as shown in Table 2. The effect estimates of being in the obesity cohort compared to controls on the odds of school completion did however not differ between the different SES indicators (OR 0,57-0,54 in Model 1, 2 and 3 respectively (Table 2)) and the added value of the composite SES variable is not clearly presented. Please discuss. Answer: Thank you for this reflection. We use a rich set of socioeconomic indicators instead of one, which we believe is a strength. SES is complex and it is plausible that certain domains of SES are more or less associated with a specific outcome. After providing specific ORs for each SES domain we can hypothesize that additive, or perhaps synergistic effects, of different SES domains may affect the outcome. Nevertheless, the OR for the childhood obesity cohort vs. comparison group remained within a narrow interval. This may indicate that there is an overlap between the measured SES domains and/or that the effect of obesity is a robust variable, independently of SES. If the referee or editor would like us to add this analysis of Table 2 in the discussion, we will be happy to do so.

Answer: Thank you for the suggestion. We have added some of this to the discussion. Please see line 357-366.

Reviewer: 2

Reviewer Name: Anders Hjern

Institution and Country: Karolinska institutet, Sweden Competing interests 1: None declared

Comments to the Author The authors have addressed most of my previous concerns adequately. However, one concern remains.

1. The authors have included anxiety/depression as a confounder in the regression analysis, which is problematic since it is probably more of a mediator/modifier. At least, that's what the authors have suggested recently in a recent article (Lindberg et al BMC Med 2020). This can be expected to lead to an underestimation of the association between obesity and education in this study. Either the data should be reanalysed without this variable or discussed in the limitations section.

Answer: Thank you for this important reflection. If e.g. Model 1 in table 2 is reanalyzed according to the suggestion above, the OR for case vs. comparison group would be 0.55 [0.49-0.61]. Hence the result is very similar. Nevertheless, we have added a sentence about this in the limitation. Please see line 363-366.